# Oxygen provision to severely ill COVID-19 patients at the peak of the 2020 pandemic in a Swedish district hospital

Anna Hvarfner[1,2]*, Ahmed Al-Djaber[3,4], Hampus Ekström[3,4], Malin Enarsson[3,5], Markus Castegren[3,6], Tim Baker[2,7◉], Carl Otto Schell[2,3◉]

1 Center for Clinical Research Dalarna, Uppsala University, Falun, Sweden, 2 Department of Global Public Health, Karolinska Institute, Stockholm, Sweden, 3 Center for Clinical Research Sörmland, Uppsala University, Eskilstuna, Sweden, 4 Department of Medicine, Nyköping Hospital, Nyköping, Sweden, 5 Department of Infectious diseases, Eskilstuna Hospital, Eskilstuna, Sweden, 6 CLINTEC and FyFa, Karolinska Institute, Stockholm, Sweden, 7 Department of Clinical Research, London School of Hygiene and Tropical Medicine, London, United Kingdom

◉ These authors contributed equally to this work.
* anna.hvarfner@gmail.com

**Data Availability Statement:** Data cannot be shared publicly because of ethical and legal restrictions linked to the content of sensitive patient information. Anonymized data can be requested via

## Abstract

Oxygen is a low-cost and life-saving therapy for patients with COVID-19. Yet, it is a limited resource in many hospitals in low income countries and in the 2020 pandemic even hospitals in richer countries reported oxygen shortages. An accurate understanding of oxygen requirements is needed for capacity planning. The World Health Organization estimates the average flow-rate of oxygen to severe COVID-19-patients to be 10 l/min. However, there is a lack of empirical data about the oxygen provision to patients. This study aimed to estimate the oxygen provision to COVID-19 patients with severe disease in a Swedish district hospital. A retrospective, medical records-based cohort study was conducted in March to May 2020 in a Swedish district hospital. All adult patients with severe COVID-19 –those who received oxygen in the ward and had no ICU-admission during their hospital stay–were included. Data were collected on the oxygen flow-rates provided to the patients throughout their hospital stay, and summary measures of oxygen provision calculated. One-hundred and twenty-six patients were included, median age was 70 years and 43% were female. On admission, 27% had a peripheral oxygen saturation of ≤91% and 54% had a respiratory rate of ≥25/min. The mean oxygen flow-rate to patients while receiving oxygen therapy was 3.0 l/min (SD 2.9) and the mean total volume of oxygen provided per patient admission was 16,000 l (SD 23,000). In conclusion, the provision of oxygen to severely ill COVID-19-patients was lower than previously estimated. Further research is required before global estimates are adjusted.

## Introduction

The world has been tackling the COVID-19 pandemic since early 2020 [1]. The pandemic has led to high demands on health systems and many lives have been lost because of the disease

Carl Otto Schell (Department of Internal medicine, Nyköping hospital, S-61185 Nyköping, Sweden; carl.schell@ki.se) or via Jesper Sperber (Head of department, Anaesthesia and Intensive care, Eskilstuna hospital, S-633 49 Eskilstuna, Sweden, Jesper.sperber@regionsormland.se) for researchers who provide a justified and reasonable research plan for which the data are required.

**Funding:** This work was supported by the Regional Research Council in Mid Sweden grant number RFR-931271. The funders had no role in the design of the study.

**Competing interests:** I have read the journal's policy and the authors of this manuscript have the following competing interests: Dr. Baker reports a grant and personal fees from Wellcome Trust, and personal fees from UNICEF and the World Bank, all outside the submitted work. This does not alter our adherence to PLOS ONE policies on sharing data and materials. The other authors declare no competing interests.

[2]. The severe acute respiratory syndrome coronvarius-2 (SARS-CoV-2) causing COVID-19 can affect most organs but is primarily a respiratory virus [3]. Viral pneumonia is the most common serious manifestation and can result in hypoxemia and acute respiratory distress syndrome (ARDS) [3,4].

To treat severe and critical COVID-19, much global focus has been on expanding intensive care capacity including the use of mechanical ventilation [5]. Whilst many critically ill patients can benefit from mechanical ventilation if it is administrated by experienced personnel, it is a staff intensive measure and requires training to be effective and avoid harm [6].

Oxygen therapy is a low cost treatment, less complex than mechanical ventilation and saves lives in COVID-19 [4,7–9]. It is the first-line treatment of hypoxemia and has been listed as a World Health Organization (WHO) essential medicine [10]. Oxygen is a limited resource in many hospitals in low income countries [11–15], and during the peaks of the pandemic waves there have been reports of hospitals running out of oxygen in high and middle income countries such as the UK, USA, South Africa, Portugal, Egypt and Brazil [16–22]. In addition, sudden failures of the oxygen systems requiring emergency transport from other sites may happen to hospitals anywhere in the world.

Capacity planning is needed to optimise the distribution of oxygen and reach highest positive impact on patient outcomes. This requires an accurate understanding of oxygen requirements for patients with COVID-19. The WHO estimates the average flow-rate of oxygen to severely ill COVID-19-patients (referring to those requiring oxygen but not intensive care unit treatment) to be 10 l/min [23]. This estimate is not based on empirical findings and there is a lack of quantitative data on oxygen provision to patients. This study aims to estimate oxygen provision to severely ill COVID-19 patients in a Swedish district hospital.

## Method

A retrospective, electronic medical records-based cohort study in the medical department in Nyköping Hospital, Sweden.

### Setting

Nyköping Hospital is a first-line district hospital in Sörmland Region in Sweden with a catchment area of 90,000 people. The medical department in the hospital has a usual capacity for managing 35 inpatients at a time. Sörmland was one of the first Swedish regions to be significantly affected by COVID-19 in 2020 [24]–the first COVID-19 patient was admitted to the department in early March. Less than a month later, the number of in-patients with COVID-19 was 43, with an average of seven new COVID-19 admissions per day.

In the medical department, 40 beds on two new wards were opened within two weeks of the first COVID-19-admission and extra staff were drawn from other areas of the hospital. Piped oxygen was provided from a central supply source to the wards. The patients' respiratory rate and peripheral oxygen saturation (SpO2) were assessed at least every two hours and any oxygen therapy through nasal prongs (<5l/min) or face masks (≥5l/min) (or, when indicated, with reservoir bags) was adjusted to maintain SpO2 within an individualized target range. The standard target range was 92–96%, but individualized lower targets ranges were used for patients with assumed hypercapnic respiratory failure. The hospital has an intensive care unit (ICU), which was expanded from the usual 5-beds to 14-beds during the pandemic. There was no shortage of oxygen during the study period.

Before rapid polymerase chain reaction (PCR) testing capacity was widely available, an initial diagnosis of COVID-19 was based on the clinical picture together with typical findings on a thoracic computed tomography (CT) scan [25]. Often the initial diagnosis was confirmed by

later PCR-testing. This early diagnostic reliance on CT scans shifted towards PCR-testing during the time of the study.

Standard treatments protocols evolved during the study period and included at different times chloroquine, antibiotics and anticoagulation prophylaxis. Corticosteroids, Remdesivir and other COVID-therapeutics were not included as standard treatment during the time of the study. High-flow nasal oxygen and non-invasive ventilation were neither recommended for ARDS in COVID-19, nor available in the medical wards at this time. Patients that needed more respiratory support than low-flow oxygen were transferred to the ICU where most received invasive mechanical ventilation.

Charlson´s age adjusted comorbidity score (CACI) was calculated for all patients and noted in the chart for each patient. The CACI score (from 0 to 37) predicts 10-year-survival in patients, accounting for age and multiple comorbidities. For example a CACI-score of 4 points estimates 53% 10-year-survival; 0 points 98% and ≥7 points 0% 10-year-survival [26,27]. As many of the admitted patients were elderly and frail, in which ICU care may cause harm or would not be in the patient´s best interest, a policy was introduced to make patient-centred decisions for every admitted patient about the appropriateness of care-escalation to ICU in the event of clinical deterioration. The decision of no escalation of care to ICU (no-ICU) was documented in the patient´s notes and, importantly, was not regarded as synonymous with end-of-life or palliative care. Patients for whom a no-ICU decision had been made received all other therapies, including oxygen therapy when indicated, unless an additional clinical decision was made to provide end-of-life palliative care.

## Study cohort

The Sörmland Region database of COVID-19 patients was used to identify participants. All adult patients (age ≥18) admitted to the department of medicine in Nyköping Hospital from March 13 to May 10 who had been diagnosed with COVID-19 during their admission and who fulfilled the criteria for "severe" disease were included. Severe patients were those who received oxygen at some point during their care in the ward and had no ICU-admission during their hospital-stay, in-line with the WHO's classification [23]. Non-severe COVID-19 patients–patients with either moderate disease (never received oxygen) or critical disease (admitted to the ICU, either from the wards or directly from the emergency department) [23]–were not included in the study cohort. However, to describe the study cohort in context, data about admission findings, the most advanced mode of respiratory support and outcomes were collected for all COVID-19 patients admitted to the department during the study period. For patients with more than one admission, only data from the first admission were included.

**Sub-groups.** Two a-priori defined sub-groups were analysed, as it was hypothesised that their oxygen provision may differ substantially from other patients. The first group were those patients for whom a no-ICU-decision had been made. The second group consisted of patients younger than 70 years old.

## Data extraction

Data were extracted in two ways. Data on vital signs and oxygen treatment on admission, patient characteristics including Charlson´s Age Adjusted Comorbidity score (CACI) [26,27], clinical and laboratory findings, pharmaceutical treatments and the presence of a no-ICU decision were extracted manually from the patients' electronic medical records. Other data were collected through a computerised search in the electronic medical records system, including for dates of admission, discharge, outcomes, and all registered SpO2 values and oxygen flow-rates throughout the patients' care. The data extracted with a computerised search were

validated by manual cross-checking a 10% sample of the data with the medical records. Data were used from the patients' entire stay in the medical department. The patients' outcome in the medical department were noted as either transfer to another department, discharge home or died, with an additional outcome of died within 60 days.

## Analysis

All data were anonymised before analysis. Oxygen provision to each patient was estimated by multiplying the oxygen flow-rate during each assessment of the patient (l/min) and the time since the previous assessment (minutes). Thereby, the volumes of oxygen provided to the patient over time were generated, and the sum of these made up the total volume of oxygen provided to the patient (formula 1).

*Formula* 1, *Total volume of oxygen provided to a patient* :

$$\sum_n X_n(T_{(n+1)} - T_n)$$

**n:** one time point in a series of consecutive time points, **$T_n$:** time at the time point n, **$X_n$:** oxygen flow (l/min) at the time point n

The mean oxygen flow-rate for each patient while receiving oxygen was calculated as the total volume of oxygen provided divided by time spent receiving oxygen treatment. An additional analysis of the mean oxygen flow-rate for each patient during their care in the ward used the total oxygen volume provided divided by total time in the ward. Means were used to provide an estimate of the total amount of oxygen that needs to be supplied to such a patient cohort. Medians were also calculated for both analyses to provide an estimate of a "typical" patient, given the non-normally distributed data. For other variables, median and interquartile range (IQR) were used for non-normally distributed values and mean and standard deviation (SD) for normally distributed values. For all admitted patients, the clinical progression score was determined as has been recently defined [28]. Admission vital signs were analysed as the proportion of vital signs corresponding to a red NEWS-2 parameter [29]. Missing descriptive data were delt with by pairwise deletion, i.e. all available data per variable was included in analysis. STATA IC/15.1(StataCorpLLC) was used for the analysis.

## Ethical considerations

The study was approved by the Ethical Review Board in Lund, reference number 2020–04012. As the study did not alter patient care and data were anonymised before analysis, individual patient consent was waived.

## Results

### Patient characteristics

In total, 206 COVID-19 patients were admitted to the department during the study period (Tables 1 and S1). Of these, 126 had severe disease and were included as the study cohort. The median age was 70 years (IQR 57–82), 54 (43%) were female and 42% had a body mass index (BMI) of 30 or above. CACI scores were 4 or above (indicating <50% estimated 10-years survival) in 50% of the patients [26,27]. A no-ICU decision was documented for 48% of the patients. On admission, 34 (27%) had an SpO2 of 91% or below and 68 (54%) had a respiratory rate of 25 or above. The length of stay was a median of 4.9 days (IQR 2.8–7.8). Eight (6.4%) patients were transferred to another department. The in-hospital mortality was 26% and the

**Table 1. Patient characteristics and outcomes.**

| | Study cohort % (n/N), unless otherwise stated | All patients admitted to the department % (n/N), unless otherwise stated |
|---|---|---|
| **Age (years), median (IQR)** | 70 (57–82) | 65 (54–78) |
| **Female** | 43% (54/126) | 42% (87/206) |
| **Diagnosis of COVID-19 confirmed by PCR** | 90% (114/126) | 89% (184/206) |
| **BMI ≥30** | 42% (38/91) | 40% (56/141) |
| **CACI ≥4** | 50% (63/126) | 43% (88/206) |
| **No-ICU-decision documented** | 48% (60/126) | 37% (77/206) |
| *Red NEWS-2* [29] *parameter on first measurements of vital signs* | | |
| **SpO2 (≤91%)** | 27% (34/126) | 28% (57/206) |
| **Respiratory rate (≤8 or ≥25 breaths/min)** | 54% (68/126) | 49% (100/203) |
| **Heart rate (≤40 or ≥131 beats/min)** | 3.2% (4/125) | 2.4% (5/205) |
| **Systolic blood pressure (≤90 or ≥220 mmHg)** | 2.4% (3/125) | 2.0% (4/205) |
| **Consciousness (Non-alert)** | 13% (17/126) | 10% (20/204) |
| **Temperature (≤35.0 or ≥39.1°C)** | 14% (18/125) | 16% (32/205) |
| *Treatments during hospital-stay* | | |
| **Antibiotics** | 79% (99/126) | 77% (158/206) |
| **Chloroquine** | 12% (15/126) | 14% (28/206) |
| **Anticoagulants** | 52% (65/126) | 55% (114/206) |
| *Patient outcomes* | | |
| **Length of stay (days), median (IQR)** | 4.9 (2.8–7.8) | 4.3 (2.2–9.0) |
| **Transfer to another department** | 6.4% (8/126) | 13% (27/206) |
| **Dead in-hospital** | 26% (33/126) | 19% (40/206) |
| **Dead at 60 days** | 32% (40/126) | 23% (48/206) |

*Abbreviations: PCR: Polymerase chain reaction, BMI: Body mass index, CACI: Charlson´s age adjusted comorbidity score, ICU: Intensive care unit, SpO2: Peripheral oxygen saturation.

60-day mortality was 32%. For the subset of patients aged <70, in-hospital and 60-day mortalities were 6.5% and 8.1% respectively and for patients with a documented no-ICU-decision 55% and 65% respectively. The clinical progression scores for all admitted patients are presented in S2 Table.

## Oxygen provision

The mean oxygen flow-rate to the patients while receiving oxygen therapy was 3.0 l/min (SD 2.9) and the median was 2.0 l/min (IQR 1.3–3.5). Results for oxygen provision to the study cohort are shown in Table 2 and for patients that were initially cared for in the wards but later transferred to ICU (therefore not part of the study cohort) in S3 Table. The highest oxygen flow-rate provided to the patients was median 4.0 l/min (IQR 2.0–8.0) and mean 5.4 l/min (SD 4.1) (Fig 1).

## Discussion

We have found that a mean of 3.0 l/min (median 2.0 l/min) oxygen were provided to severely ill COVID-19 patients while receiving oxygen in a Swedish district hospital during the first

**Table 2. Oxygen provision.**

| | Study cohort (n = 126) | Subgroups | |
|---|---|---|---|
| | | Patients aged <70 (n = 62) | Patients with a no-ICU-decision (n = 60) |
| **Age, median (IQR)** | 70 (57–82) | 57 (48–61) | 83 (75–88) |
| **Days on oxygen treatment, median (IQR)** | 2.3 (0.68–4.2) | 1.8 (0.68–3.9) | 2.5 (0.77–4.8) |
| **Oxygen flow to patients during oxygen therapy (l/min)** | | | |
| • Mean (SD) | 3.0 (2.9) | 2.6 (2.3) | 4.0 (3.7) |
| • Median (IQR) | 2.0 (1.3–3.5) | 1.9 (1.3–2.9) | 2.9 (1.6–5.6) |
| **Oxygen flow to patients during time in the ward (l/min)** | | | |
| • Mean (SD) | 2.2 (2.9) | 1.8 (2.3) | 3.1 (3.7) |
| • Median (IQR) | 1.2 (0.32–2.6) | 1.1 (0.43–2.0) | 1.6 (0.28–4.6) |
| **Total volume of oxygen provided per patient admission (l)** | | | |
| • Mean (SD) | 16,000 (23,000) | 12,000 (16,000) | 22,000 (26,000) |
| • Median (IQR) | 7,400 (1,200–21,000) | 4,8000 (1,800–17,000) | 12,000 (2,000–32,000) |

peak of the pandemic in 2020. This calculated oxygen flow-rate is lower than the 10 l/min estimated by the WHO for COVID-19 patients with severe disease [23].

A strength of this study is the use of a method for calculating oxygen provision to hospitalized patients by summing up documented flow-rates in the medical records. Previously, findings of the proportions of patients receiving oxygen [8,30–33] and assumptions on required oxygen flow [23] have been presented. To the best of our knowledge, this study is the first to use all the necessary data to be able to calculate total oxygen provision to patients.

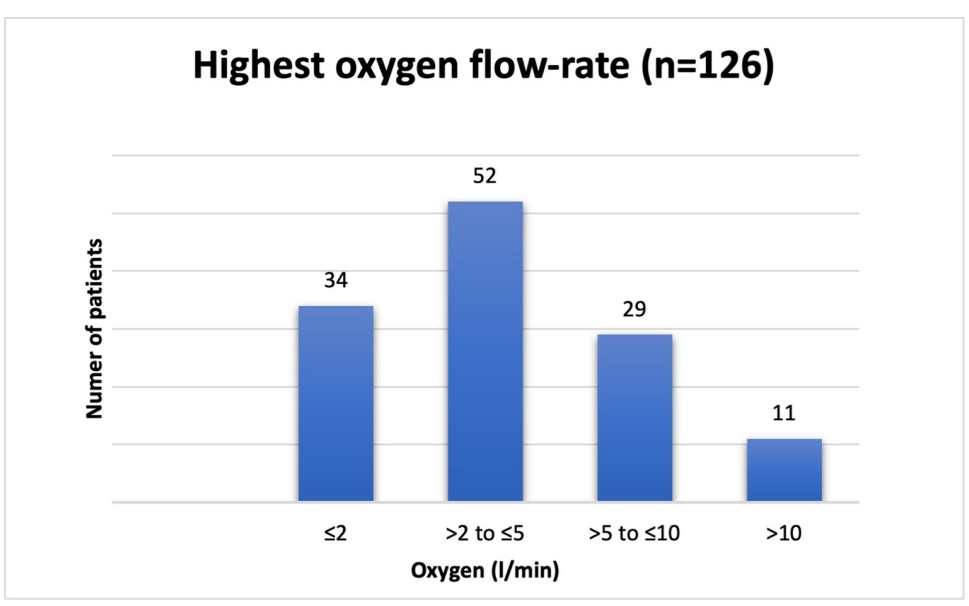

**Fig 1. Highest oxygen flow-rate.** The figure shows the highest oxygen flow-rate provided to patients with severe COVID-19 during their care in the medical wards.

The 206 patients admitted to the medical department in Nyköping with COVID-19 during this study period had similar age, gender proportions and in-hospital mortality as other described cohorts of hospitalised COVID-19 patients [8,30,32,34]. The median length of stay of 4.3 days was shorter in the Nyköping patient group than in two cohorts from New York and Madrid described during the early pandemic, as well as in the multinational ISARIC survey, with stays of seven to nine days [32,34,35]. This could be explained by Sweden´s early discharge policies [36,37] and a considerable amount of patient transfers in this study (13% of all admitted patients and 6.4% of the study cohort) due to overcrowding.

The lower flow-rates of oxygen to patients with severe COVID-19 in this study compared to previous estimates are interesting but, for several reasons, should be interpreted with caution when planning health care services. Firstly, it is treacherous for health care systems to make plans based on the COVID-19 patient categories "severe" and "critical" as there are currently several definitions in use and many of them categorise patients based on the care they receive rather than the care they require [23,28,38]. Since resources and practices for ICU-admission vary greatly around the world, for example ICU beds per 100,000 population vary from 29 in Germany to 6 in Sweden and <1 in Uganda [39], very few COVID-19 patients would be classified as critical in Uganda according to the definition used by WHO and in this study. Indeed, in settings with less availability of ICU beds, sicker patients will be classified as "severe" and require higher flows of oxygen than the cohort in this study. In addition, suboptimal respiratory support might cause shorter survival or survival with prolonged time to recovery–hence it is difficult to make assumptions on the quantity of oxygen they need without available data. This study aimed to study a patient group that was truly severe according to the WHO classification, and due to the well-resourced hospital system in Sweden succeeded with this aim, but a similar group may not be easy to delineate in other settings.

Secondly, decisions on treatment limitations were important to the assignment of patients to the severe group in this study and such practices vary largely across countries. Of the study patients, 48% had a no-ICU-decision meaning that even if they deteriorated, they were kept in the general wards and received targeted oxygen treatment. Although oxygen needs for this subgroup was also considerably lower than the WHO estimate (4.0 l/min), it is possible that in settings with other norms around treatment limitations, these patients may, at some point, have received ICU-treatments such as mechanical ventilation with substantially higher oxygen flow-rates.

Thirdly, the flow-rates of oxygen provided to patients depend on target saturation and duration between treatment modifications. The patients in this study were cared for with defined targets and frequent saturation controls. In wards where oxygen flow is not–or cannot be–adjusted as frequently, oxygen flow-rates and target saturations may be higher to provide patients with a safe margin for avoiding hypoxia [40]. While the optimal target saturation for hospitalised patients receiving oxygen is debated [41,42], the target range in Nyköping was set following the Surviving Sepsis Campaign guidelines [43].

Our findings suggest that the oxygen need for severely ill COVID-19 patients may be lower than previously estimated. Future research using the methods described in this study in larger cohorts and from other settings would be useful to inform capacity planning with updated estimates of oxygen need. Additionally, as oxygen is essential in many other conditions such as sepsis, trauma [15,22,44–46] and notably child pneumonia, a disease killing 800,000 children under 5 each year [14,47], estimates for the oxygen needs for treating these conditions would be beneficial.

## Conclusion

The provision of oxygen to severely ill COVID-19-patients was lower than previously estimated. Further research is required before global estimates are adjusted.

## Supporting information

**S1 Table. Patient characteristics and outcomes.**
(DOCX)

**S2 Table. Clinical progression score for all admitted patients.**
(DOCX)

**S3 Table. Oxygen provision in the wards to patients that were later transferred to ICU.**
(DOCX)

## Acknowledgments

The authors would like to thank Johan Printz and Josefin Back for conducting the computerised search in the electronic medical records system, Karin Näslund for her general IT-support throughout the study and the heads of the medical department in Nyköping, Björn Alpe and Karin Marminge, for their support of this study.

## Author Contributions

**Conceptualization:** Tim Baker, Carl Otto Schell.

**Data curation:** Anna Hvarfner, Ahmed Al-Djaber, Hampus Ekström, Malin Enarsson.

**Formal analysis:** Anna Hvarfner, Tim Baker, Carl Otto Schell.

**Investigation:** Anna Hvarfner, Ahmed Al-Djaber, Hampus Ekström.

**Methodology:** Anna Hvarfner, Ahmed Al-Djaber, Hampus Ekström, Markus Castegren, Tim Baker, Carl Otto Schell.

**Project administration:** Anna Hvarfner.

**Supervision:** Tim Baker, Carl Otto Schell.

**Validation:** Anna Hvarfner, Ahmed Al-Djaber, Hampus Ekström, Carl Otto Schell.

**Writing – original draft:** Anna Hvarfner, Tim Baker, Carl Otto Schell.

**Writing – review & editing:** Anna Hvarfner, Ahmed Al-Djaber, Hampus Ekström, Malin Enarsson, Markus Castegren, Tim Baker, Carl Otto Schell.

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
