## [Decision Letter · Decision Letter 0]

19 Oct 2021

PONE-D-21-10716

Oxygen provision to severely ill COVID-19 patients at the peak of the 2020 pandemic in a Swedish district hospital

PLOS ONE

Dear Dr. Hvarfner,

Thank you for submitting your manuscript to PLOS ONE. After careful consideration, we feel that it has merit but does not fully meet PLOS ONE’s publication criteria as it currently stands. Therefore, we invite you to submit a revised version of the manuscript that addresses the points raised during the review process.

Please revise accordingly.

We look forward to receiving your revised manuscript.

Kind regards,

Academic Editor

PLOS ONE

Journal Requirements:

[I have read the journal's policy and the authors of this manuscript have following competing interests: Dr. Baker reports a grant and personal fees from Wellcome Trust, and personal fees from UNICEF and the World Bank, all outside the submitted work.].

Reviewers' comments:

Reviewer's Responses to Questions

**Comments to the Author**

1. Is the manuscript technically sound, and do the data support the conclusions?

Reviewer #1: Yes

Reviewer #2: Yes

2. Has the statistical analysis been performed appropriately and rigorously? 

Reviewer #1: Yes

Reviewer #2: Yes

3. Have the authors made all data underlying the findings in their manuscript fully available?

Reviewer #1: Yes

Reviewer #2: Yes

4. Is the manuscript presented in an intelligible fashion and written in standard English?

Reviewer #1: Yes

Reviewer #2: Yes

5. Review Comments to the Author

Reviewer #1: The study does seem to be interesting especially in this pandemic where a lot of countries are facing paucity of oxygen.

Few comments though.

1)Could the authors please elaborate on the CACI scoring system for the benefit of the readers?

2)Did any of the patients have any underlying pulmonary disease like Bronchial asthma,COPD, structural lung disease like bronchiectasis?

3)Did any of them have underlying cardiac issues which could have aggravated the hypoxemia?

4)How was 02 administered ,nasal prongs,face masks,Venturi masks etc?

5)What was the source of 02 supply-concentrators, cylinders,central supply source?

Would be great to hear from you.

Regards,

Dr Supriya.

Reviewer #2: Thank you for allowing me to review " Oxygen provision to severely ill COVID-19 patients at the peak of the 2020 pandemic in a Swedish district hospital" by Anna Hvarfner et al.

The authors retrospectively surveyed oxygen flow-rate and total volume of oxygen in severely ill COVID-19-patients. The mean oxygen flow-rate was 3.0 l/min, and lower than previously estimated. This is an interesting study. Although oxygen flow-rate might vary according to targeted SpO2, the authors stated about it.

My comments are as follows.

“The WHO estimates the average flow-rate of oxygen to severely ill COVID 19-patients (referring to those requiring oxygen but not intensive care unit treatment) to be 10 l/min.”

I couldn't find this description in reference 22. Where is it listed?

There seems to some missing data in table 1. How do you handle missing values?

6. PLOS authors have the option to publish the peer review history of their article (what does this mean?). If published, this will include your full peer review and any attached files.

Reviewer #1: No

Reviewer #2: No

---

## [Author Response · Author response to Decision Letter 0]

3 Dec 2021

REVIEWER #1

The study does seem to be interesting especially in this pandemic where a lot of countries are facing paucity of oxygen. Few comments though. 

 - Thank you!

1)Could the authors please elaborate on the CACI scoring system for the benefit of the readers? 

 - The following is added in rows 108-119: 

Charlson´s age adjusted comorbidity score (CACI) was calculated for all patients and noted in the chart for each patient. The CACI score (from 0 to 37) predicts 10-year-survival in patients, accounting for age and multiple comorbidities For example a CACI-score of 4 points estimates 53% 10-year-survival; 0 points 98% and �7 points 0% 10-year-survival(26,27). 

2)Did any of the patients have any underlying pulmonary disease like Bronchial asthma,COPD, structural lung disease like bronchiectasis? 

 - We did not collect data on pre-existing diagnoses in detail but chose to use the CACI score as an indicator of the total burden of underlying illness for each patient.

3)Did any of them have underlying cardiac issues which could have aggravated the hypoxemia? 

 - Please, see above. 

4)How was 02 administered ,nasal prongs,face masks,Venturi masks etc?

 - We have added this description on O2-administration in rows 92-94:

“(…)oxygen therapy through nasal prongs (<5l/min) or face masks (≥5l/min) (or, when indicated, with reservoir bags) was adjusted to maintain SpO2 within an individualized target range.”

5)What was the source of 02 supply-concentrators, cylinders,central supply source?

 - All oxygen came from a central supply source. We have added this in rows 90-91:

”Piped oxygen was provided from a central supply source to the wards.”

REVIEWER#2: Thank you for allowing me to review " Oxygen provision to severely ill COVID-19 patients at the peak of the 2020 pandemic in a Swedish district hospital" by Anna Hvarfner et al.

The authors retrospectively surveyed oxygen flow-rate and total volume of oxygen in severely ill COVID-19-patients. The mean oxygen flow-rate was 3.0 l/min, and lower than previously estimated. This is an interesting study. Although oxygen flow-rate might vary according to targeted SpO2, the authors stated about it.

My comments are as follows.

 - Thank you!

“The WHO estimates the average flow-rate of oxygen to severely ill COVID 19-patients (referring to those requiring oxygen but not intensive care unit treatment) to be 10 l/min.”

I couldn't find this description in reference 22. Where is it listed?

 - We agree that this is not very clearly stated in the reference. However in the first paragraph of the “COVID-19 and oxygen” section there is a description of different severity stages of COVID-19 stating that “…about 15 % of them have severe illness requiring oxygen therapy, and 5% will be critically ill requiring intensive care unit treatment.” Further down in table 1 there is a presentation of the oxygen needs for these two groups (for severe patients 10 L/min). 

There seems to some missing data in table 1. How do you handle missing values? 

 - We used pairwise deletion, i.e.. including all available cases per variable in the descriptive data in Table 1. We have added a sentence clarifying this in rows 181-182. 

“Missing descriptive data were delt with by pairwise deletion, i.e. all available data per variable was included in analysis.”

Journal Requirements and our actions

 - To the best of our ability the title page, the main body, and the supporting information have been adjusted. The figure file is relabeled. 

 2.Please review your reference list to ensure that it is complete and correct. If you have cited papers that have been retracted, please include the rationale for doing so in the manuscript text, or remove these references and replace them with relevant current references. Any changes to the reference list should be mentioned in the rebuttal letter that accompanies your revised manuscript. If you need to cite a retracted article, indicate the article’s retracted status in the References list and also include a citation and full reference for the retraction notice.

 - The reference list is reviewed. We have added one reference to a recently published study of the unmet need of oxygen treatment in critical illness (15).

[I have read the journal's policy and the authors of this manuscript have following competing interests: Dr. Baker reports a grant and personal fees from Wellcome Trust, and personal fees from UNICEF and the World Bank, all outside the submitted work.].

 - Done, see cover letter and here: 

I have read the journal's policy and the authors of this manuscript have the following competing interests: Dr. Baker reports a grant and personal fees from Wellcome Trust, and personal fees from UNICEF and the World Bank, all outside the submitted work. This does not alter our adherence to PLOS ONE policies on sharing data and materials. The other authors declare no competing interests. 

 - Updated data availability statement added in cover letter and here: 

Data cannot be shared publicly because of ethical and legal restrictions linked to the content of sensitive patient information. Anonymized data can be requested via Carl Otto Schell (Department of Internal medicine, Nyköping hospital, S-61185 Nyköping, Sweden; carl.schell@ki.se) or via Jesper Sperber (Head of department, Anaesthesia and Intensive care, Eskilstuna hospital, S-633 49 Eskilstuna, Sweden, Jesper.sperber@regionsormland.se) for researchers who provide a justified and reasonable research plan for which the data are required. 

- Done

---

## [Decision Letter · Decision Letter 1]

17 Dec 2021

PONE-D-21-10716R1Oxygen provision to severely ill COVID-19 patients at the peak of the 2020 pandemic in a Swedish district hospitalPLOS ONE

Dear Dr. Hvarfner,

Thank you for submitting your manuscript to PLOS ONE. After careful consideration, we feel that it has merit but does not fully meet PLOS ONE’s publication criteria as it currently stands. Therefore, we invite you to submit a revised version of the manuscript that addresses the points raised during the review process.

Please revise.

We look forward to receiving your revised manuscript.

Kind regards,

Academic Editor

PLOS ONE

Reviewers' comments:

Reviewer's Responses to Questions

**Comments to the Author**

1. If the authors have adequately addressed your comments raised in a previous round of review and you feel that this manuscript is now acceptable for publication, you may indicate that here to bypass the “Comments to the Author” section, enter your conflict of interest statement in the “Confidential to Editor” section, and submit your "Accept" recommendation.

Reviewer #1: All comments have been addressed

Reviewer #2: All comments have been addressed

2. Is the manuscript technically sound, and do the data support the conclusions?

Reviewer #1: Yes

Reviewer #2: Yes

3. Has the statistical analysis been performed appropriately and rigorously? 

Reviewer #1: Yes

Reviewer #2: Yes

4. Have the authors made all data underlying the findings in their manuscript fully available?

Reviewer #1: Yes

Reviewer #2: Yes

5. Is the manuscript presented in an intelligible fashion and written in standard English?

Reviewer #1: Yes

Reviewer #2: Yes

6. Review Comments to the Author

Reviewer #1: (No Response)

Reviewer #2: Thank you for allowing me to review " Oxygen provision to severely ill COVID-19 patients at the peak of the 2020 pandemic in a Swedish district hospital" by Anna Hvarfner et al.

The authors addressed the reviewer’s comments on a point-by-point basis.

My comments are as follows.

“- We agree that this is not very clearly stated in the reference. However in the first

paragraph of the “COVID-19 and oxygen” section there is a description of different

severity stages of COVID-19 stating that “…about 15 % of them have severe illness

requiring oxygen therapy, and 5% will be critically ill requiring intensive care unit

treatment.” Further down in table 1 there is a presentation of the oxygen needs for

these two groups (for severe patients 10 L/min).”

Sorry, I could not find the authors description in reference 23. Where is it listed? I could not find table 1.

7. PLOS authors have the option to publish the peer review history of their article (what does this mean?). If published, this will include your full peer review and any attached files.

Reviewer #1: No

Reviewer #2: No

---

## [Author Response · Author response to Decision Letter 1]

23 Dec 2021

REVIEWER´S COMMENT

Reviewer #2: Thank you for allowing me to review " Oxygen provision to severely ill COVID-19 patients at the peak of the 2020 pandemic in a Swedish district hospital" by Anna Hvarfner et al.

The authors addressed the reviewer’s comments on a point-by-point basis. My comments are as follows.

“- We agree that this is not very clearly stated in the reference. However in the first

paragraph of the “COVID-19 and oxygen” section there is a description of different

severity stages of COVID-19 stating that “…about 15 % of them have severe illness

requiring oxygen therapy, and 5% will be critically ill requiring intensive care unit

treatment.” Further down in table 1 there is a presentation of the oxygen needs for

these two groups (for severe patients 10 L/min).”

Sorry, I could not find the authors description in reference 23. Where is it listed? I could not find table 1.

OUR RESPONSE

We are sorry that our previous response was not clear. Reference 23 contains the information in the first paragraph of the “COVID-19 and oxygen” section on the first page and in Table 1 on the second page. We highlight the relevant parts in screenshots in the attached file "Response to Reviewers".

---

## [Decision Letter · Decision Letter 2]

5 Jan 2022

Oxygen provision to severely ill COVID-19 patients at the peak of the 2020 pandemic in a Swedish district hospital

PONE-D-21-10716R2

Dear Dr. Hvarfner,

We’re pleased to inform you that your manuscript has been judged scientifically suitable for publication and will be formally accepted for publication once it meets all outstanding technical requirements.

Kind regards,

Academic Editor

PLOS ONE

Additional Editor Comments (optional):

Reviewers' comments:

Reviewer's Responses to Questions

**Comments to the Author**

1. If the authors have adequately addressed your comments raised in a previous round of review and you feel that this manuscript is now acceptable for publication, you may indicate that here to bypass the “Comments to the Author” section, enter your conflict of interest statement in the “Confidential to Editor” section, and submit your "Accept" recommendation.

Reviewer #1: All comments have been addressed

Reviewer #2: All comments have been addressed

2. Is the manuscript technically sound, and do the data support the conclusions?

Reviewer #1: Yes

Reviewer #2: Yes

3. Has the statistical analysis been performed appropriately and rigorously? 

Reviewer #1: Yes

Reviewer #2: Yes

4. Have the authors made all data underlying the findings in their manuscript fully available?

Reviewer #1: Yes

Reviewer #2: Yes

5. Is the manuscript presented in an intelligible fashion and written in standard English?

Reviewer #1: Yes

Reviewer #2: Yes

6. Review Comments to the Author

Reviewer #1: (No Response)

Reviewer #2: Thank you for allowing me to review " Oxygen provision to severely ill COVID-19 patients at the peak of the 2020 pandemic in a Swedish district hospital" by Anna Hvarfner et al. The authors addressed the reviewer’s comments on a point-by-point basis.

7. PLOS authors have the option to publish the peer review history of their article (what does this mean?). If published, this will include your full peer review and any attached files.

Reviewer #1: No

Reviewer #2: No

---

## [Editor Report · Acceptance letter]

10 Jan 2022

PONE-D-21-10716R2 

Oxygen provision to severely ill COVID-19 patients at the peak of the 2020 pandemic in a Swedish district hospital 

Dear Dr. Hvarfner:

I'm pleased to inform you that your manuscript has been deemed suitable for publication in PLOS ONE. Congratulations! Your manuscript is now with our production department. 

Kind regards, 

on behalf of

Dr. Robert Jeenchen Chen 

Academic Editor

PLOS ONE